# Lung Ultrasound and Neutrophil Lymphocyte Ratio in Early Diagnosis and Differentiation between Viral and Bacterial Pneumonia in Young Children

**DOI:** 10.3390/children9101457

**Published:** 2022-09-23

**Authors:** Ahmed Omran, Heba Awad, Mostafa Ibrahim, Sonya El-Sharkawy, Samar Elfiky, Ahmed R. Rezk

**Affiliations:** 1Department of Pediatrics & Neonatology, Faculty of Medicine, Suez Canal University, Ismailia 41522, Egypt; 2Department of Radiodiagnosis, Faculty of Medicine, Suez Canal University, Ismailia 41522, Egypt; 3Department of Pediatrics & Neonatology, Faculty of Medicine, Port Said University, Port Said 42526, Egypt; 4Department of Pediatrics, Ain Shams University, Cairo 11566, Egypt

**Keywords:** bacterial pneumonia, viral pneumonia, lung ultrasound, infant

## Abstract

Lung ultrasound (LUS) is a crucial diagnostic tool for identifying pneumonia in the pediatric age group. However, it plays a limited role in the early distinction between viral and bacterial pneumonia in children. The objectives of our study were to determine if LUS and the neutrophil-lymphocyte ratio (NLR) were useful in identifying and distinguishing between viral and bacterial pneumonia in Egyptian children under the age of two. Within the first 12 h of being admitted to our department, 52 children with clinical symptoms and signs suggestive of community-acquired pneumonia (CAP) underwent LUS and the NLR. LUS and the NLR strongly differentiated children with viral from those with bacterial pneumonia. For the early diagnosis and differentiation between viral and bacterial pneumonia in young Egyptian children, LUS was proven to be a noninvasive and reliable method. Combining the NLR with LUS increased the diagnostic accuracy when evaluating children suspected of having pneumonia.

## 1. Introduction

Pediatric pneumonia is a leading cause of morbidity and mortality worldwide and remains a diagnostic challenge, especially in developing countries [1]. In less than five years, childhood pneumonia accounts for 18% of all deaths worldwide. Traditionally, the diagnosis of pneumonia is based on the three pillars: physical examination, including auscultation, laboratory evaluation and imaging studies.

Chest radiography (CXR) is important for diagnosing pneumonia, but in clinical practice, an early differentiation between viral and bacterial pneumonia in children is very difficult [2]. Bacterial pneumonia cannot be distinguished from nonbacterial pneumonia using the CXR [3,4]. Diagnostic and treatment delays can lead to a significant increase in mortality, while the inappropriate use of antibiotics to treat nonbacterial respiratory infections contributes to the development of antibiotic resistance.

Lung ultrasound (LUS) has emerged as a very important non-ionizing radiation diagnostic tool in the detection of lung and pleural diseases in pediatrics [5,6,7,8,9,10]. LUS has shown high diagnostic accuracy compared to CXR and computed tomography (CT) in the diagnosis and follow-up of community-acquired pneumonia (CAP) in pediatrics [11,12]. Furthermore, it successfully differentiates pneumonia from other diseases that cause respiratory distress (RD) in pediatrics [13].

The neutrophil-lymphocyte ratio (NLR) is a novel inflammatory biomarker that can be used to detect systemic inflammation. It is defined as the absolute neutrophil count/absolute lymphocyte count. Several studies investigated the role of the NLR in the diagnosis, follow-up, and prognosis of neonatal and childhood pneumonia [14,15,16,17,18].

The aim of this study was to clarify the role of LUS and the NLR in the early detection and differentiation of viral and bacterial pneumonia in young Egyptian children.

## 2. Material and Methods

### 2.1. Patient’s Characteristics

The study protocol was approved by the Institutional Review and Research Ethics Committee of the School of Medicine, reference number (2850#), and this prospective study was conducted from October 2016 to July 2017. Children who were referred to a pediatric emergency department and who met the following inclusion criteria were included: (1) 2 months to 2 years of age; (2) clinical symptoms and signs suggestive of CAP (fever, cough, tachypnea, decreased and/or crackling breath sounds); (3) CXR performance, according to the clinical judgment of the pediatrician; (4) the availability of a sonographer for the LUS performance during the patient’s initial evaluation; (5) obtaining written informed consent from the parent for the child’s participation in the study.

The exclusion criteria were (1) children referred to our hospital with complicated pneumonia; (2) patients with major cardiac anomalies or with chronic pulmonary diseases; (3) children who were unstable hemodynamically; (4) clinical diagnosis of bronchiolitis; and (5) patients with unresolved pneumonia or receiving treatment for pneumonia within the preceding month.

At the beginning of the study, children were evaluated by clinical examination, which included their medical history, clinical examination, C-reactive protein (CRP) levels, and complete blood count, including NLR, sputum culture and PCR for respiratory virus testing.

The sputum culture samples were obtained from children on two consecutive mornings by early morning gastric aspiration. Gastric lavage was performed after an overnight fast of at least 4 h, as described by Zar et al. [19].

In children, pneumonia was preliminarily diagnosed by two senior pediatricians based on the diagnostic criteria for pneumonia set out in the revised British Thoracic Society (BTS) guidelines [20]. Sixty children hospitalized with an initial diagnosis of pneumonia underwent CXR and LUS within the first 12 h after hospitalization.

### 2.2. CXR Examination

The children were subjected to a posterior-anterior CXR view in the supine position using the same equipment. Lateral radiographs were not obtained in accordance with the BTS guidelines. The radiologist was blinded to the child’s clinical presentation. 

### 2.3. LUS Examination

A transthoracic LUS examination was performed using a high-resolution linear transducer (Philips HD7) with frequencies between 7 and 12 MHz. In a single session, the LUS evaluation was performed in real-time. The same senior expert sonographer performed all LUS examinations, which were blinded to the results of CXR.

The chest was divided into three sections: lateral, anterior, and posterior, to cover the entire lung surface. All areas were scanned in the transverse and longitudinal planes, superior-inferior and medial-lateral. With the child in the supine position, the anterolateral area was examined. The posterior region was examined in the prone position, according to Copetti et al. [21].

The LUS findings were performed as described by Caiulo et al. [22]. Classification: (1) normal lung pattern with or without A-lines, (2) presence of focal multiple or confluent B-lines, (3) abnormal pleural lines, presenting as irregular pleural lines, and (4) the presence of subpleural lung consolidation, with or without air bronchogram.

### 2.4. Statistical Analysis

The continuous variables are expressed as the mean ± standard deviation. The categorical variables are displayed as counts and percentages. All statistical analyses were performed using the SPSS/PC software package, version 20 (SPSS Inc, Chicago, IL, USA). Numerical data were compared using the Student’s *t*-test, and categorical data were compared using the chi-square (χ^2^) or Fisher’s exact test. For all tests, *p*-values < 0.05 were considered statistically significant.

## 3. Results

### 3.1. Patient Characteristics 

Between October 2016 and July 2017, a total of 60 children were hospitalized with a lower respiratory tract infection (LRTI) and underwent CXR and LUS within their first 12 h after admission; then, based on their symptoms, a clinical examination, laboratory investigations, including virological and bacteriological results, and radiological investigations, including the CXR and LUS results, were classified into 34 cases (65.3%) having bacterial pneumonia, 18 cases (34.6%) having viral pneumonia and 8 cases were excluded from the study (Figure 1). Ten patients had bacterial pneumonia with a negative sputum culture based on predominant neutrophilia, an elevated CRP level, and CXR findings, including (dense lobar consolidations and alveolar consolidations with air bronchograms).The demographic, clinical, and laboratory findings in the children with viral and bacterial pneumonia are presented in (Table 1).

### 3.2. CXR and LUS Findings

In bacterial pneumonia, the CXR showed consolidation patches in 24 cases (70.5%), which was statistically significant (*p*-value = 0.002). On the other hand, in viral pneumonia, there was a predominance of reticulonodular opacities and increased bronchovascular markings in 8 cases (38.8%), which was statistically significant (*p*-value 0.02 and 0.04, respectively). LUS for bacterial pneumonia showed subpleural consolidation greater than 1 cm (85.2%), and an air bronchogram (88.8%) being the most common features, with statistically significant results (*p*-values 0.0001 and <0.0001, respectively). Other LUS findings included pleural line irregularities and decreased lung sliding, which were also statistically significant (*p*-value, 0.009 and< 0.0001), respectively. In viral pneumonia, multiple B-lines (83.8%) with various numbers and involving different levels of intercostal spaces was a statistically significant finding (*p*-value, <0.0001) and, along with subpleural consolidations <1 cm (77.8%), were the prominent findings, as seen in (Table 2) and (Figure 2).

### 3.3. Pneumonic Patch Site and Size and CXR Findings

It was noticed that 5 (10%) of the cases in both groups had negative CXR findings despite positive LUSs. This is probably related mainly to the patch’s size and/or site, as most of the cases were found to have the patch in the posterior region and/or were less than <1cm in size (Figure 3).

### 3.4. Diagnostic Performances of LUS, CXR and NLR 

LUS showed higher sensitivity and specificity than the CXR in diagnosing bacterial pneumonia, with LUS and CXR having sensitivities of 88.2% and 70.6% and specificities of 100% and 85.7%, respectively. LUS was also highly sensitive in regards to the diagnosis of viral pneumonia compared to CXR, with a sensitivity of 83.3% and 33.3% for LUS and CXR, respectively; it is also more specific, with a specificity of 85.7% and 78.5 % for LUS and CXR, respectively. The NLR showed sensitivity and specificity (73.5% and 94.4%), respectively, in the diagnosis of bacterial pneumonia (Table 3). According to the ROC curves shown in (Figure 4), the sensitivity of both CXR and LUS increased when combined with the NLR. 

### 3.5. Sensitivity, Specificity, PPV and NPV for Each CXR and LUS Findings

Regarding the diagnostic performance of the individual CXR findings, the highest sensitivity was with the increased bronchovascular markings (82.8%) and reticulonodular opacity (100 %), and the highest specificity was found with consolidation (87.5%) and the air bronchogram (93.5%). As for the individual LUS finding, the highest sensitivity was with B-lines (83.3%), followed by subpleural consolidation (88.2%) and air bronchograms (88.2%), and the highest specificity was for air bronchograms (94.4%) followed by reduced lung sliding (88.9%) and B-lines (88.2%) (Table 4).

## 4. Discussion

Infectious pneumonia in pediatrics is still considered a highly prevalent and morbid problem worldwide. In Egypt, pneumonia and other acute LRTIs probably account for 10% of deaths in children under the age of 5 [23].The presenting features of viral and bacterial pneumonia in children are nearly similar. Early and accurate differentiation between viral and bacterial pneumonia in children is, therefore, a major challenge for pediatricians in routine practice.

This study evaluated the role of LUS and the NLR in the early detection and differentiation of viral and bacterial pneumonia in young Egyptian children.

Ideally, the etiology of pneumonia should be determined by isolating the causative pathogens [24]. In our study, sputum cultures from 70.6% of children who had been diagnosed with bacterial pneumonia were positive. In cases when a culture is negative, we rely on high neutrophil counts, elevated CRP levels, and CXR findings, which are highly suggestive of bacterial pneumonia. In order to address the diagnostic challenge in children with sputum culture-negative bacterial pneumonia, Huang et al. rely on elevated TLC and CRP as indirect indicators of bacterial infection in 40.77% of their bacterial pneumonia cases with a negative sputum culture [25]. Furthermore, Ogawa H et al., in their meta-analysis, reported that bacterial pathogens were detected in only 73% of good-quality sputum cultures, which is very close to our results [26].

In our study, CXR did not identify 5 (10%) cases of pneumonia in both groups, which instead were detected by LUS. Radiographic failures are primarily related to the size of the pneumonic patch (<1 cm) and its posterior location. Ho et al. reported that CXR showed no obvious lesions in 12 (7.4%) patients [10], and Iorio et al. also reported that 4 (7.6%) had negative CXR findings [27]. Lung consolidation of less than 1 cm has been reported to be undetectable by CXR [28]. They explained the failure of CXR in lesion detection due to (1) the small size and radiolucency in the early stages of the pneumonic process, (2) the posterior position of the lesion beyond the heart or mediastinum, and (3) the alveolar pattern of the pneumonic patch [10,27,28].

In our study, LUS was superior to CXR in diagnosing bacterial and viral pneumonia. In children with bacterial pneumonia, LUS had a sensitivity, specificity, NPV and PPV of 88.2%, 100%, 77.8%, and 100%, respectively, compared with CXR, which had a lower sensitivity, specificity, NPV and PPV of 70.6%, 85.7%, 54.5%, and 92.3%, respectively. In children with viral pneumonia, the LUS showed a sensitivity, specificity, NPV and PPV of 83.3%, 85.7%, 80% and 88.2%, respectively, compared to CXR, which showed a lower sensitivity, specificity, NPV and PPV of 33.3%, 78.5%, 47.8% and 66.7%, respectively. These differences between the two groups may be attributed to the size of the pneumonic patches, which were smaller in the cases of viral pneumonia.

This issue was addressed by Claes et al., who found that both LUS and CXR lacked specificity in detecting small areas of lung consolidation [6]. In pediatrics, it is very difficult to distinguish the areas of consolidation from small areas of focal atelectasis due to bronchial obstruction and inflammation [6].

Consistent with our results, many investigators have reported a higher diagnostic accuracy of LUS compared with CXR in diagnosing CAP in children [5,6,7,9,29]. Pereda et al., in their meta-analysis, the diagnostic performance of LUS in children had a sensitivity of 96% (95% CI, 94–97%) and a specificity of 93% (95% CI, 90–96%), and was found in the diagnosis of CAP [30].

In our study, we found that the LUS findings were different in children with bacterial and viral pneumonia. In bacterial pneumonia, irregular pleural lines were observed in 79.5% compared to 38.8% in viral pneumonia. Lung patch sizes that were greater than 1 cm and air bronchograms were more common in children with bacterial pneumonia than in children with viral pneumonia (88.2%/22.2%) and (88.2%/5.5%). The presence of multiple B-lines and normal lung slides were more prominent in infants with viral pneumonia than in those with bacterial pneumonia (83.8%/11.4% and 88.8%/23.5%, respectively).

Other researchers found that a subpleural consolidation larger than 2 cm is predictive of bacterial pneumonia, while one between 0.5 and 2 cm is probably suggestive of viral pneumonia in children. Additionally, they noted that LUS had a higher sensitivity and specificity in identifying bacterial pneumonia in children as opposed to viral pneumonia [31,32].

Other investigators also reported that the presence of small subpleural consolidation and/or B-lines or confluent B-lines without air bronchograms was considered an ultrasound finding associated with viral pneumonia or bronchiolitis [8,33,34,35,36]. In bacterial pneumonia, greater pulmonary consolidation with sonographic air bronchograms is the most prominent feature [33]. LUS also has the potential to diagnose the nature of lung effusion and distinguish between bacterial and viral pneumonia [33,36].

Tsung et al. defined positive or negative patients with viral pneumonia according to the presence of small subpleural consolidations (usually less than 0.5 cm) and/or single or confluent B-lines, and patients were defined as positive or negative for bacterial pneumonia according to the presence or absence of lung consolidation with air bronchograms [33]. Berg et al. reported that LUS is a sensitive method for detecting pediatric pneumonia and also helps distinguish between bacterial, viral, and atypical pneumonia, mainly depending on the infiltrate size. The infiltrate size was 3.9 cm for bacterial pneumonia, 3.3 cm for atypical pneumonia, and 1.8 cm for viral pneumonia [37].

In our study, the most sensitive LUS findings in diagnosing pneumonia were subpleural consolidation and air bronchogram in children, with a sensitivity of 88.2% for both. The estimated sensitivity was very close to previously reported studies, with a sensitivity range of 86–100% [12,28,29].

The NLR has proven to be a simple and useful marker of systemic inflammation. The reported normal values for the NLR are highly variable in neonatal, pediatric and adult populations [14,38,39,40,41]. In our study, the NLR mean was 1.8 in children with bacterial pneumonia compared to 0.66 in children with viral pneumonia. The difference was statistically significant (*p* = 0.0001). The overall sensitivity, specificity, NPV and PPV of the NRP were 73.5%, 94.4%, 65.4%, and 96.1%, respectively. Interestingly, Bekdas et al. showed that the NLR is useful for diagnosing viral and bacterial pneumonia and predicting complications in pediatrics [18]. Recently, the NLR was provided as an independent predictor of mortality and a worse outcome in COVID-19 patients [42].

No previous studies have examined the combination of the NLR and CXR or LUS for the diagnosis of pneumonia in children. In this study, the combination of the NLR and CXR or LUS improved the sensitivity of diagnosing pneumonia in childrencompared to the sensitivity of the NLR, CXR and LUS alone.

Nazerian et al. showed that the LUS sensitivity in combination with procalcitonin for diagnosing pneumonia in children was significantly higher than that of LUS and/or PCT alone [43].

In our study, distinguishing between viral and bacterial pneumonia reduced the use of antibiotics for viral pneumonia. Jones et al. reported that LUS’s ability to detect subcentimeter lung consolidations could increase the rate of antibiotic treatment and may increase the rate of hospital admission [8]. Tsung et al. concluded that the visualization of viral pneumonia on LUS in pandemic cases or epidemic influenza patients might be useful in making immediate decisions to initiate empiric treatment with antiviral medication [33].

Our study has several notable limitations. Firstly, the relatively small number of included children. Secondly, lack of follow-up to confirm improvement or deterioration. Thirdly, we did not perform chest CT scans on our patients and did not take it as the gold-standard test to diagnose pneumonia due to the high cost and the risk of radiation exposure. Finally, all LUS scans were performed by senior trained radiologists. This raises questions about the broad applicability of this technique, especially for inexperienced examiners.

## 5. Conclusions

LUS has been found to be a noninvasive and accurate method for early detection and differentiation of viral and bacterial pneumonia in young children. The combination of the NLR and LUS increased the diagnostic accuracy when evaluating children with suspected pneumonia.

## Figures and Tables

**Figure 1 children-09-01457-f001:**
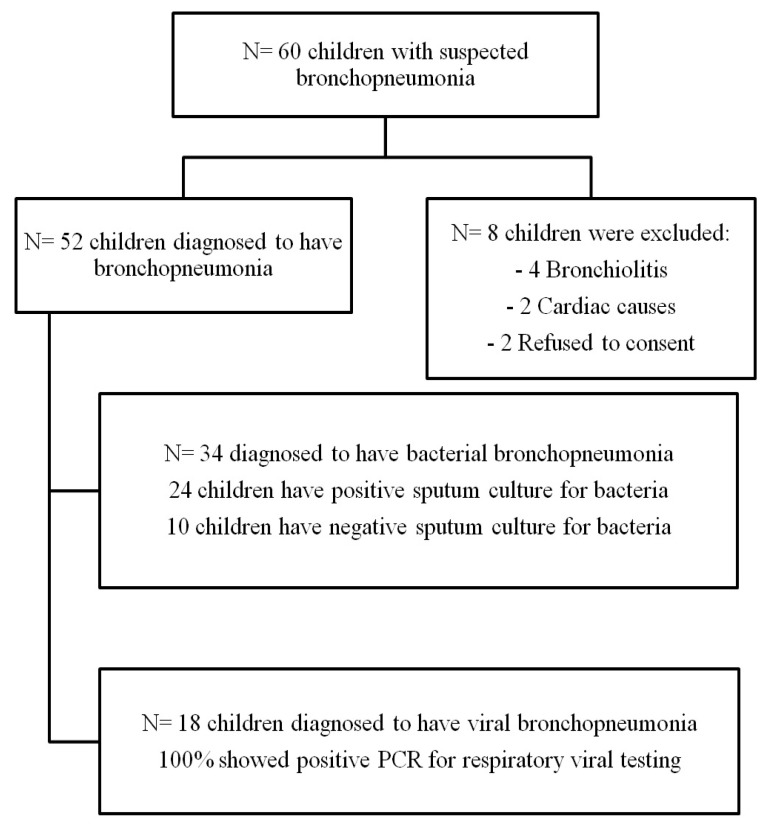
Patients flow chart.

**Figure 2 children-09-01457-f002:**
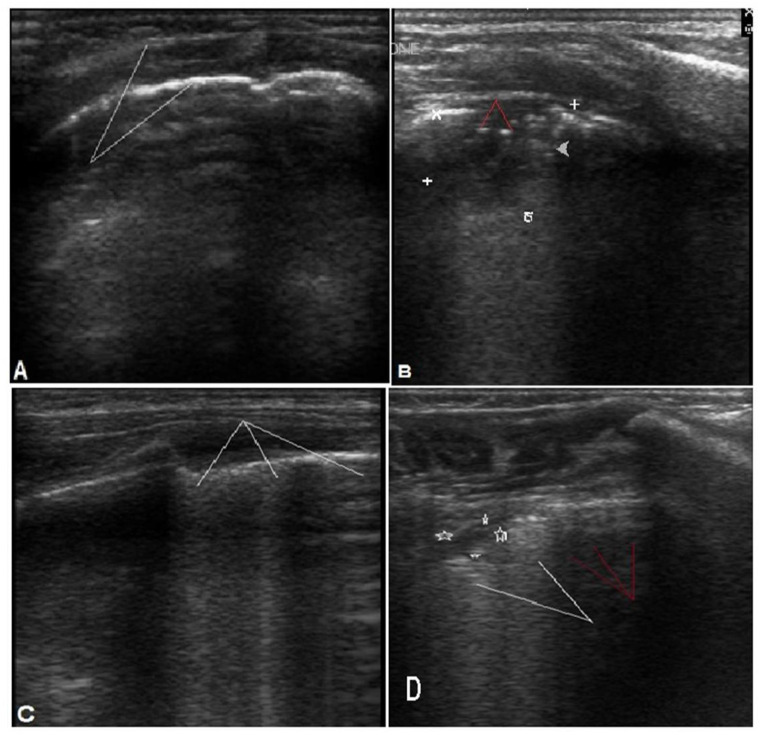
Different LUS findings in bacterial and viral pneumonia. (**A**,**B**) for bacterial pneumonia: (**A**) shows irregular pleural line (arrows). (**B**): consolidation 1.6 × 2 cm (white marks) with air bronchogram (red arrows). (**C**,**D**) for viral pneumonia. (**C**): multiple (**B**) lines. (**D**): shows subpleural consolidation <1 cm (marks) with multiple confluent (white arrows) and scattered (red arrows) B lines.

**Figure 3 children-09-01457-f003:**
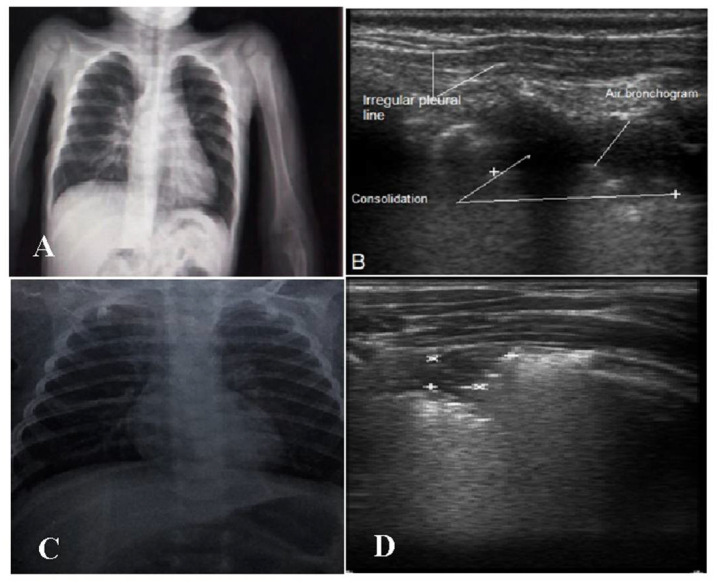
(**A**,**B**) Two-year-old male with bacterial bronchopneumonia. (**A**): CXR showing no consolidation. (**B**): LUS with consolidation patch, air bronchogram and irregular pleural line. (**C**,**D**) Eight-month-old male with viral pneumonia. (**C**): Normal CXR. (**D**): Small subpleural consolidation 0.6 × 0.8 cm (marks) in the right upper lung region posteriorly.

**Figure 4 children-09-01457-f004:**
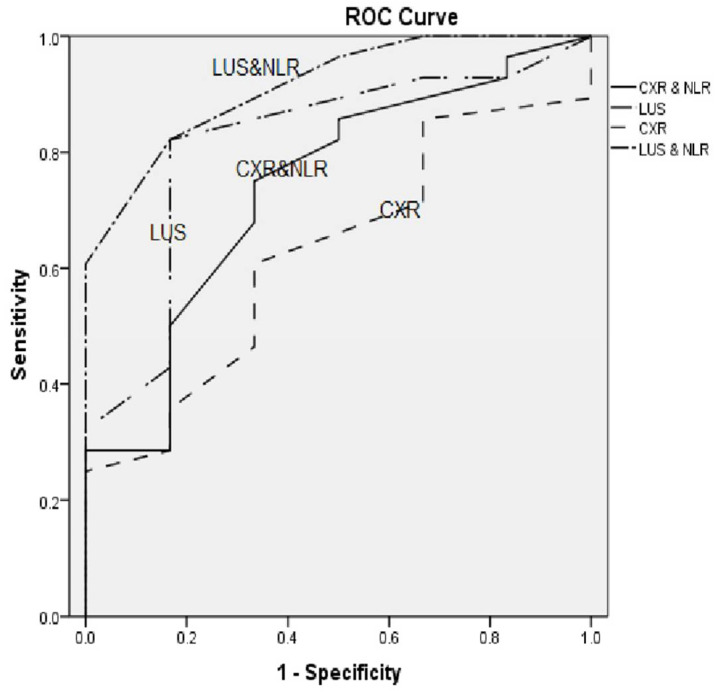
ROC curves for CXR, CXR + NLR, LUS and LUS + NLR.

**Table 1 children-09-01457-t001:** Baseline patient characteristics.

		Bacterial Pneumonia (*n* =34) Mean ± SD	Viral Pneumonia (*n* =18) Mean ± SD	*p*-Value
**Demographic data**				
Age (Months)		7.4± 6.8	12± 8.3	0.03 *
Gender	Male	14 (41.2%)	10 (55.5%)	0.5
	Female	20 (58.8%)	8 (44.5%)	
Weight (Kg)		7.6± 3.5	8.8± 2.2	0.19
Average hospital stay (days)		5.7±2.3	4.5±2.1	0.07
**Complaint**				
Cough		30 (88.2%)	10 (55.5%)	0.02 *
Fever > 38		25 (73.5%)	8 (44.4%)	0.07 *
RD		25 (73.5%)	16 (88.8%)	0.8
Feeding difficulties		20 (58.8%)	15 (83.3%)	0.13
**Chest examination**				
Diminished air entry		30 (88.2%)	4 (22.2%)	<0.0001 *
Fine crepitation		29 (85.2%)	7 (38.8%)	<0.0001 *
Wheezes		11 (32.3%)	16 (88.8%)	0.0003 *
**Laboratory findings**				
TLC (mean)		14,000/cmm	8000/cmm	<0.0001 *
NLR (mean)		1.8	0.66	<0.0001 *
CRP (mean) **Positive sputum culture** **PCR for respiratory viral testing**		13 mg/L 24 (70.5%) 0 (0%)	5 mg/L 0 (0%) 18 (100%)	<0.0001 * <0.0001 * <0.0001 *

RD: respiratory distress; TLC: total leucocytic count; NLR: neutrophil/lymphocyte ratio; CRP: C-reactive protein; PCR: polymerase chain reaction. * *p* < 0.05.

**Table 2 children-09-01457-t002:** CXR and LUS findings in both groups.

		Bacterial Pneumonia (*n* =34) Mean ± SD	Viral Pneumonia (*n* =18) Mean ± SD	*p*-Value
**CXR Findings**				
Increased bronchovascular markings		6 (17.6%)	8 (38.8%)	0.04 *
Consolidation	Present	24 (70.5%)	4 (22.2%)	0.002 *
	Absent	10 (29.5%)	14 (77.8%)	
Reticulonodular opacities		0 (0%)	8 (38.8%)	0.02 *
Air bronchogram		11 (32.3%)	2 (11.1%)	0.17
**LUS Findings**				
Pleural line	Regular	7 (20.5%)	11 (61.2%)	0.009 *
	Disturbed	27 (79.5%)	7 (38.8%)	
Subpleural consolidation	>1 cm	30 (88.2%)	4 (22.2%)	<0.0001 *
	<1 cm	4 (11.8%)	14 (77.8%)	
Air bronchogram	Yes	30 (88.2%)	1 (5.5%)	<0.0001 *
	No	4 (11.8%)	17 (94.4%)	
B-lines	Yes	4 (11.8%)	15 (83.8%)	<0.0001 *
	No	30 (88.2%)	3 (16.2%)	
Lung sliding	Normal	8 (23.5%)	16 (88.8%)	<0.0001 *
	Decreased	26 (76.4%)	2 (11.2%)	

* *p* < 0.05.

**Table 3 children-09-01457-t003:** Diagnostic performances of LUS, CXR and NLR.

		Sensitivity (95% CI)	Specificity (95% CI)	NPV (95% CI)	PPV (95% CI)	+LR (95% CI)	−LR (95% CI)
LUS	Bacterial Pneumonia	88.2 (72.5–96.7%)	100 (76.8–100%)	77.8 (58.2–89.8%)	100.00	0.0	0.12 (0.05–0.3)
	Viral Pneumonia	83.3 (85.6–96.4%)	85.7 (57.1–98.2%)	80 (58.2–92%)	88.2 (67.1–96.4%)	5.8 (1.6–21.4)	0.19 (0.07–0.56)
CXR	Bacterial Pneumonia	70.6(52.2–84.9%)	85.7(57.1–98.2%)	54.5(40.6–67.8%)	92.3(76.5–97.8%)	4.9 (1.3–18.1)	0.34(0.2–0.6)
	Viral Pneumonia	33.3 (13.3–59%)	78.5 (49.2–95.3%)	47.8 (37.4–58.4%)	66.7 (37.7–86.9%)	1.5 (0.5–5.1)	0.85 (0.55–1.30)
**Bacterial pneumonia**							
NLR		73.5(55.6–87.1%)	94.4(72.7–99.7%)	65.4(51.6–77%)	96.1(78.6–99.4%)	13.24 (1.95–89.87)	0.28(0.16–0.50)

**Table 4 children-09-01457-t004:** Sensitivity, specificity, PPV and NPV for each CXR and LUS finding.

	Sensitivity (95% CI)	Specificity (95% CI)	NPV (95% CI)	PPV (95% CI)	+LR (95% CI)	−LR (95% CI)
**CXR Findings**						
Increased bronchovascular markings	82.8(66.3–93.4%)	38.8(17.3–64.2%)	53.8 (31.5–74.7%)	72.5 (63.9–79.7%)	1.36 (0.91–2.02)	0.44 (0.17–1.12)
Consolidation	70.5 (52.5–84.9%)	87.5 (71–96.5%)	73.7 (62–82.7%)	85.7 (71.1–93.6%)	5.6 (2.2–14.4)	0.34 (0.21–0.67)
Reticulonodular opacities	100 (89.7–100%)	22.2 (6.4–47.6%)	100%	70.8 (65.4–75.6%)	1.29 (1.00–1.65)	0.00
Air bronchogram	47.8 (26.8–69.4%)	93.5 (79.1–99.2%)	71.4 (62.6–78.9%)	84.6 (57.3–95.7%)	7.5 (1.8–31.2)	0.56 (0.37–0.83)
**LUS Findings**						
Pleural line	79.4 (62.1–91.3%)	61.1 (35.7–82.7%)	61.1 (42.4–77%)	79.4 (67.8–87.5%)	2.04 (1.12–3.74)	0.34 (0.16–0.72)
Subpleural consolidation	88.2 (72.5–96.7%)	77.7 (52.3–93.5%)	77.7 (57.4–90%)	88.2 (75.8–94.7%)	3.97 (1.66–9.51)	0.15 (0.06–0.39)
Air bronchogram	88.2 (72.5–96.7%)	94.4 (72.7–99.9%)	80.9 (62.7–91.5%)	96.8 (81.6–99.5%)	15.9 (2.35–107.1)	0.12 (0.05–0.31)
B-lines	83.3 (58.6–96.4%)	88.2 (72.5–96.7%)	90.9 (77.9–96.6%)	78.9 (59.3–90.6%)	7.08 (2.76–18.2)	0.19 (0.07–0.53)
Lung sliding	76.5 (58.8–89.2%)	88.9 (65.3–98.6%)	66.7 (51.6–78.9%)	92.9 (77.6–98%)	6.88 (1.84–25.76)	0.26 (0.14–0.50)

## Data Availability

Datasets are available on request.

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
