# Peer review of "Lung Ultrasound and Neutrophil Lymphocyte Ratio in Early Diagnosis and Differentiation between Viral and Bacterial Pneumonia in Young Children"

_children, 2022, doi:10.3390/children9101457_

Round 1

Reviewer 1 Report

The paper by Omran et al aimed to detect the role of LUS and NLR in early detection and differentiation between viral and bacterial pneumonia in young Egyptian children. The authors showed that combining NLR to LUS increase the diagnostic accuracy when evaluating infant suspected of having pneumonia.

The paper is well written.

Minor review

1.      Can you add protocol number of your study?

2.      Can you include limitation of your study?

Author Response

Reviewer 1

The paper by Omran et al aimed to detect the role of LUS and NLR in the early detection and differentiation between viral and bacterial pneumonia in young Egyptian children. The authors showed that combining NLR with LUS increase the diagnostic accuracy when evaluating infant suspected of having pneumonia.

The paper is well written.

Response:

Thank you for your valuable time in reviewing our manuscript. We appreciate your positive comments. We believe correcting our manuscript according to your kind suggestion markedly improved it.

  1. Reviewer’s Comments:

Can you add the protocol number of your study?

Author's Response: The study protocol number was added

  1. Reviewer’s Comments:

Can you include limitations of your study?

Author's Response: we added the limitations in our study

Our study has several notable limitations. Firstly, the relatively small number of included children. Secondly, lack of follow-up to confirm improvement or deterioration. Thirdly, we did not perform chest CT scans on our patients and did not take it as the gold-standard test to diagnose pneumonia due high cost and the risk of radiation exposure. Finally, all LUS scans were performed by senior trained radiologists. This raises questions about the broad applicability of this technique, especially for inexperienced examiners.

Reviewer 2 Report

1.- It would be interesting that you explain about the NLR :

     Definition, the normal values, the Abnormal values, the cutoff point in          order to differentiate Bacterial vs Viral Pneumonia.

And then you can speak about Sensitivity, Specificity, etc

2.- I recommend you, to read these papers:

     a.- Adil Elabbas et al.

Lung Ultrasonography Beyond the Diagnosis of Pediatrics Pneumonia.

Cureus 2022 Feb; 14(2): e22460

     b.- Matteo Reggolo, Mauro Vaccaro, Alessandra Sorce, Benedetta Stancanelli, Michele Colaci, Giuseppe Natoli, Mario Russo, Innocenza Alessandria.

       Neutrophil to Lymphocyte Ratio (NLR) is a promising predictor of mortality and Admission to Intensive Care Unit of COVID 19 patients.

      Journal of Clinical Medicine 2022, 11, 2235

In order to consider them, in the Discussion.

Author Response

Thank you for your valuable time in reviewing our manuscript. We appreciate your positive comments. We believe that correcting our manuscript according to your kind suggestion markedly improved it.

Reviewer comment:

1.- It would be interesting that you explain about the NLR :

     Definition, the normal values, the Abnormal values, the cutoff point in          order to differentiate Bacterial vs Viral Pneumonia.

And then you can speak about Sensitivity, Specificity, etc

Authors response:

We added in the introduction part and discussion the points requested by the reviewer.

Reviewer comments 

2.- I recommend you, to read these papers:

     a.- Adil Elabbas et al.

Lung Ultrasonography Beyond the Diagnosis of Pediatrics Pneumonia.

Cureus 2022 Feb; 14(2): e22460

     b.- Matteo Reggolo, Mauro Vaccaro, Alessandra Sorce, Benedetta Stancanelli, Michele Colaci, Giuseppe Natoli, Mario Russo, Innocenza Alessandria.

       Neutrophil to Lymphocyte Ratio (NLR) is a promising predictor of mortality and Admission to Intensive Care Unit of COVID 19 patients.

      Journal of Clinical Medicine 2022, 11, 2235

In order to consider them, in the Discussion.

Authors response:

Thank you very much for your kind suggestion.

We added all your suggestions to the discussion of our manuscript.

Reviewer 3 Report

Dear authors,

thank you for the interesting paper dealing with a topic of high relevance in pediatric every da life.

Unfortunately I do not 100% understand the methodology and how the results were figured out.

These are my questions/recommendations:

1) Figure 1: 
    a) What exactly were the criteria to classify into bacterial or viral pneumonia? Was this classification done before or after obtaining virological/bacteriological results? How could you decide that 10 pneumonias were bacterial without positive culture results? When criteria were Sono und Rö, später nicht als Beweis nutzbar, dass es das war...
   b) How could you obtain sputum samples in children younger than two years of age?
   c) Did you additionally find virus in children classified as "bacterial pneumonia" or bacteria in those classified as "viral pneumonia"?

Overall, methodology, assessment and evaluation is not clear for me. I understood that you defined pneumonias as viral or bacterial using radiological examination results which you later evaluated as helpful for discrimination between those entities. Here the black cat bites into its own tail -or not? 

Author Response

Reviewer 2

Reviewer’s Comment:

Thank you for the interesting paper dealing with a topic of high relevance in pediatric everyday life. Unfortunately, I do not 100% understand the methodology and how the results were figured out.

Author's Response: Thank you for your valuable time in reviewing our manuscript. We appreciate your positive comments. We will do our best to make our manuscript clear according to your kind suggestions.

Reviewer’s Comment:

Figure 1: 

  1. a) What exactly were the criteria to classify into bacterial or viral pneumonia? Was this classification done before or after obtaining virological/bacteriological results? How could you decide that 10 pneumonias were bacterial without positive culture results? When criteria were Sono und Rö, später nicht als Beweis nutzbar, dass es das war...

Author's Response: Thank you for your questions

1) What exactly were the criteria to classify into bacterial or viral pneumonia?

Due to the lack of specific criteria to classify into bacterial or viral pneumonia in children, the determination of pneumonia etiology is ideally based upon the isolation of causative pathogens which could be time-consuming and may be very difficult in limited recourses settings. Our study aim was to detect the role of LUS and N/L ratio in diagnosis and early differentiation between bacterial and viral pneumonia in Egyptian young children. We included in our study children presented with features of LRTI and provisionally diagnosed with pneumonia by two senior pediatricians based on the diagnostic criteria for pneumonia set out in the revised British Thoracic Society (BTS) guidelines.

Harris M, Clark J, Coote N, et al. British Thoracic Society guidelines for the management of community acquired pneumonia in children: update 2011. Thorax 2011;66::ii1–23.

2) Was this classification done before or after obtaining virological/bacteriological results?

The classifications of our patients were done based on the presenting symptoms, clinical examination, and laboratory and radiological results, they were classified into 34 cases (65.3%) having bacterial pneumonia, and 18 cases (34.6%) having viral pneumonia. This final classification and diagnosis were done after all laboratory and radiological investigations including (virological/bacteriological results).

3) How could you decide that 10 pneumonias were bacterial without positive culture results?

This depends on white blood cell (WBC) count with predominant neutrophilia, elevated CRP level, and CXR findings including (dense lobar consolidations, alveolar consolidations with air bronchograms) and negative viral test.

Ogawa H et al., in their meta-analysis, evaluated 24 studies including >4500 patients who detected bacterial pathogens in only 73% of good-quality sputum Gram stains; while detection rates dropped to 36%  when including lower quality Grain stains as well.

Ogawa H, Kitsios GD, Iwata M, Terasawa T. Sputum Gram Stain for Bacterial Pathogen Diagnosis in Community-acquired Pneumonia: A Systematic Review and Bayesian Meta-analysis of Diagnostic Accuracy and Yield. Clin Infect Dis 2020; 71:499.

4) When criteria were Sono und Rö, später nicht als Beweis nutzbar, dass es das war...

We are sorry but we can’t understand what the reviewer means.

Reviewer’s Comment:

1) How could you obtain sputum samples in children younger than two years of age?

Author's Response: Thank you for your questions

We obtained sputum samples from our patients by early morning gastric aspiration on each two consecutive mornings. Gastric lavage (GL) was performed after an overnight fast of at least four hours as performed in Zar et al.

 Zar HJ, Tannenbaum E, Apolles P, Roux P, Hanslo D, Hussey G. Sputum induction for the diagnosis of pulmonary tuberculosis in infants and young children in an urban setting in South Africa. Arch Dis Child. 2000 Apr;82(4):305-8. doi: 10.1136/adc.82.4.305. PMID: 10735837; PMCID: PMC1718283.

Reviewer’s Comment:

Did you additionally find virus in children classified as "bacterial pneumonia" or bacteria in those classified as "viral pneumonia"?
Author's Response:

None of our patients were found to have mixed infections

Round 2

Reviewer 3 Report

Dear authors,

thank you for answering my questions and improving your manuscript.

Still, my main concern is that I understand you defined pneumonias being bacterial or viral by methods that are established (BTS guidelines). Later you prove 10 pneumonias being of bacterial origin by using the same methods without positive culture results. For my understanding that is not pemitted. Please explain that. Maybe I do not understand you right. All the rest is much improved now. Still I find your topic highly interesting, helpful and suitable for publication. But that one point I find crucial to clearify.

Author Response

Reviewer comments:

thank you for answering my questions and improving your manuscript.

Still, my main concern is that I understand you defined pneumonias being bacterial or viral by methods that are established (BTS guidelines). Later you prove 10 pneumonias being of bacterial origin by using the same methods without positive culture results. For my understanding that is not pemitted. Please explain that. Maybe I do not understand you right. All the rest is much improved now. Still I find your topic highly interesting, helpful and suitable for publication. But that one point I find crucial to clearify.

Authors response:

Dear Professor,

Thank you for your valuable time in reviewing our manuscript. We appreciate your very kind comments and suggestions.

Regarding the diagnostic challenge in bacterial pneumonia cases with negative sputum culture. We added our explanation in the discussion part as follows:

Ideally, the etiology of pneumonia should be determined by isolating the causative pathogens [24]. In our study, sputum cultures from 70.6% of children who have been diagnosed with bacterial pneumonia are positive. In cases when a culture is negative, we rely on high neutrophil counts, elevated CRP levels, and CXR findings highly suggestive of bacterial pneumonia. In order to address the diagnostic challenge in children with sputum culture negative bacterial pneumonia, Huang et al. rely on elevated TLC and CRP as indirect indicators of bacterial infection in 40.77% of their bacterial pneumonia cases with negative sputum culture [25]. Furthermore, Ogawa H et al., in their meta-analysis, reported that bacterial pathogens were detected in only 73% of good-quality sputum cultures which is very close to our results [26].

24. Loens K, Van Heirstraeten L, Malhotra-Kumar S, Goossens H, Ieven M. Optimal sampling sites and methods for detection of pathogens possibly causing community-acquired lower respiratory tract infections. J Clin Microbiol. 2009;47(1):21-31.

25. Huang WY, Lee MS, Lin LM, Liu YC. Diagnostic performance of the Sputum Gram Stain in predicting sputum culture results for critically ill pediatric patients with pneumonia. Pediatr Neonatol. 2020;61(4):420-425.

26. Ogawa H, Kitsios GD, Iwata M, Terasawa T. Sputum Gram Stain for Bacterial Pathogen Diagnosis in Community-acquired Pneumonia: A Systematic Review and Bayesian Meta-analysis of Diagnostic Accuracy and Yield. Clin Infect Dis. 2020:27;71(3):499-513.

We hope that could explain our point of view.

Thank you so much.